# Radiomics for Dynamic Lung Cancer Risk Prediction in USPSTF-Ineligible Patients

**DOI:** 10.3390/cancers17213406

**Published:** 2025-10-23

**Authors:** Morteza Salehjahromi, Hui Li, Eman Showkatian, Maliazurina B. Saad, Mohamed Qayati, Sherif M. Ismail, Sheeba J. Sujit, Amgad Muneer, Muhammad Aminu, Lingzhi Hong, Xiaoyu Han, Simon Heeke, Tina Cascone, Xiuning Le, Natalie Vokes, Don L. Gibbons, Iakovos Toumazis, Edwin J. Ostrin, Mara B. Antonoff, Ara A. Vaporciyan, David Jaffray, Fernando U. Kay, Brett W. Carter, Carol C. Wu, Myrna C. B. Godoy, J. Jack Lee, David E. Gerber, John V. Heymach, Jianjun Zhang, Jia Wu

**Affiliations:** 1Department of Imaging Physics, MD Anderson Cancer Center, Houston, TX 77030, USA; msalehjahromi@mdanderson.org (M.S.); mohakayaty@gmail.com (M.Q.); amabdulraheem@mdanderson.org (A.M.);; 2Department of Thoracic/Head and Neck Medical Oncology, MD Anderson Cancer Center, Houston, TX 77030, USA; 3Institute for Data Science in Oncology, MD Anderson Cancer Center, Houston, TX 77030, USA; 4Department of Health Services Research, The University of Texas MD Anderson Cancer Center, Houston, TX 77030, USA; 5Department of General Internal Medicine, The University of Texas MD Anderson Cancer Center, Houston, TX 77030, USA; 6Department of Thoracic and Cardiovascular Surgery, The University of Texas MD Anderson Cancer Center, Houston, TX 77030, USA; 7Department of Radiology, UT Southwestern Medical Center, Dallas, TX 75235, USA; 8Department of Thoracic Imaging, The University of Texas MD Anderson Cancer Center, Houston, TX 77030, USA; bcarter2@mdanderson.org (B.W.C.);; 9Department of Biostatistics, The University of Texas MD Anderson Cancer Center, Houston, TX 77030, USA; 10Harold C. Simmons Comprehensive Cancer Center, UT Southwestern Medical Center, Dallas, TX 75235, USA; david.gerber@utsouthwestern.edu; 11Division of Hematology-Oncology, Department of Internal Medicine, UT Southwestern Medical Center, Dallas, TX 75235, USA; 12Peter O’Donnell Jr. School of Public Health, UT Southwestern Medical Center, Dallas, TX 75235, USA; 13Department of Genomic Medicine, MD Anderson Cancer Center, Houston, TX 77030, USA; 14Lung Cancer Genomics Program, MD Anderson Cancer Center, Houston, TX 77030, USA; 15Lung Cancer Interception Program, MD Anderson Cancer Center, Houston, TX 77030, USA

**Keywords:** lung cancer risk prediction USPSTF, light-smokers, pulmonary nodules, radiomics

## Abstract

**Simple Summary:**

Many people who develop lung cancer have never smoked or have smoked very little, and they are often not included in current CT screening programs. We analyzed how small lung nodules change across a series of routine CT scans and turned those changes into simple image-based measurements (“radiomics”) that capture growth, density, and texture. Our approach follows both the nodule and the surrounding lung over time, creating a personalized picture of risk. We also found that widely used tools such as the Brock model and deep learning methods designed for screening heavy smokers were less accurate in our non- and light-smoker cohort. By combining longitudinal imaging features with basic clinical information, our tool more reliably identified higher-risk patients who might otherwise be missed by today’s screening rules. These findings suggest that tracking nodules and lung context over time can support earlier evaluation and intervention for people not covered by existing screening guidelines.

**Abstract:**

**Background**: Non-smokers and individuals with minimal smoking history represent a significant proportion of lung cancer cases but are often overlooked in current risk assessment models. Pulmonary nodules are commonly detected incidentally—appearing in approximately 24–31% of all chest CT scans regardless of smoking status. However, most established risk models, such as the Brock model, were developed using cohorts heavily enriched with individuals who have substantial smoking histories. This limits their generalizability to non-smoking and light-smoking populations, highlighting the need for more inclusive and tailored risk prediction strategies. **Purpose**: We aimed to develop a longitudinal radiomics-based approach for lung cancer risk prediction, integrating time-varying radiomic modeling to enhance early detection in USPSTF-ineligible patients. **Methods**: Unlike conventional models that rely on a single scan, we conducted a longitudinal analysis of 122 patients who were later diagnosed with lung cancer, with a total of 622 CT scans analyzed. Of these patients, 69% were former smokers, while 30% had never smoked. Quantitative radiomic features were extracted from serial chest CT scans to capture temporal changes in nodule evolution. A time-varying survival model was implemented to dynamically assess lung cancer risk. Additionally, we evaluated the integration of handcrafted radiomic features and the deep learning-based Sybil model to determine the added value of combining local nodule characteristics with global lung assessments. **Results**: Our radiomic analysis identified specific CT patterns associated with malignant transformation, including increased nodule size, voxel intensity, textural entropy, as indicators of tumor heterogeneity and progression. Integrating radiomics, delta-radiomics, and longitudinal imaging features resulted in the optimal predictive performance during cross-validation (concordance index [C-index]: 0.69), surpassing that of models using demographics alone (C-index: 0.50) and Sybil alone (C-index: 0.54). Compared to the Brock model (67% accuracy, 100% sensitivity, 33% specificity), our composite risk model achieved 78% accuracy, 89% sensitivity, and 67% specificity, demonstrating improved early cancer risk stratification. Kaplan–Meier curves and individualized cancer development probability functions further validated the model’s ability to track dynamic risk progression for individual patients. Visual analysis of longitudinal CT scans confirmed alignment between predicted risk and evolving nodule characteristics. **Conclusions**: Our study demonstrates that integrating radiomics, sybil, and clinical factors enhances future lung cancer risk prediction in USPSTF-ineligible patients, outperforming existing models and supporting personalized screening and early intervention strategies.

## 1. Introduction

While smoking is widely recognized as the primary risk factor for lung cancer, it is essential to understand that nonsmokers are also at risk. Indeed, about 20% of individuals who died from lung cancer in the United States in 2018 had never smoked. Globally, lung cancer in never-smokers is the fifth leading cause of cancer-related deaths [1]. This highlights the urgent need for further research to improve our understanding of the disease in nonsmokers. Notably, the U.S. Preventive Services Task Force (USPSTF) currently does not recommend low-dose CT screening for individuals who have never smoked, underscoring a critical gap in clinical practice and research [2].

Recent advancements have provided promising tools for addressing these challenges. The Brock University model [3], for instance, represents a significant leap forward in estimating the cancer risk of pulmonary nodules identified on initial CT scans. This model leverages both radiologists’ assessments and patient demographics, improving diagnostic accuracy and assisting clinicians in formulating appropriate management strategies. Similarly, deep learning (DL) models such as “Sybil” have demonstrated potential in predicting lung cancer risk using a single low-dose computed tomography (LDCT) scan [4]. However, it is important to note that both models have been developed and validated primarily in screening populations composed of heavy smokers meeting USPSTF criteria [5], leaving uncertainty about their performance in non-screening populations, including non- and light smokers. This highlights the need for additional research tailored to these underrepresented groups. In this regard, radiomics has emerged as a powerful approach, capable of quantifying a nodule’s shape, size, texture, and intensity from CT scans [6,7] offering a potential avenue for more inclusive risk assessment.

In this study, we analyze a real-world cohort of patients from our in-house GEMINI database, all of whom were ineligible for USPSTF screening. While a proportion of this cohort comprises non- and light smokers, the study broadly aims to improve risk assessment in individuals who do not meet traditional screening criteria. Leveraging a longitudinal radiomics approach on serial CT scans, we aim to enhance early detection and risk stratification by capturing temporal changes in imaging features at multiple time points. This approach enables a more dynamic and individualized assessment of lung cancer risk, addressing a critical gap in the clinical management of these incidentally identified nodules.

## 2. Materials and Methods

### 2.1. Longitudinal Patient Data Curation

We queried the MD Anderson GEMINI database, a single-center, real-world cohort that integrates clinical, molecular, and imaging data from routine care, to construct a longitudinal cohort of patients’ ineligible for USPSTF lung cancer screening. Inclusion criteria were: (1) adults ineligible for USPSTF lung cancer screening (smoking history <20 pack-years and/or quit >15 years ago); (2). Individuals have available demographic information and CT scans.

### 2.2. Study Design

The overall aim of this study is to build a dynamic risk modeling system based on clinical and imaging data to predict the likelihood of developing lung cancer during follow-up, and further benchmark it with existing models such as the Brock and Sybil models (Figure 1b). We first extracted radiomic features of nodules on individual CT scans and computed the variations (Dradiomics) between scans at different time points. To compute cancer risk based on CT scans, we applied the Sybil model [4]. To model the effect of features that change over time on the risk of developing cancer, we adopted the time-varying survival regression technique. We also included baseline patient demographic information for risk modeling. Given the different categories of extracted features, we implemented selection strategies to preprocess them.

### 2.3. Details of Extracted Multi-Modal Features

Radiomics: Institutional radiologists performed quality checks of CT scans and manually segmented target nodules from longitudinal CT images using software such as 3D Slicer (v4.11.20200930) or ITK-SNAP 3.8.0. A total of 107 radiomic features were extracted from each primary nodule using the PyRadiomics package in Python 3.8, including 14 shape-based, 18 first-order statistical, and multiple texture-based features (24 GLCM, 14 GLDM, 16 GLRLM, 16 GLSZM, and 5 NGTDM).

DRadiomics: we calculated the difference in feature values between two consecutive scans, then normalized these differences by the time elapsed between the two scans to mitigate the irregularity of time intervals in a real world setting [8].

Sybil risk scores: we extracted cancer risk by applying the pre-trained Sybil model [4], which estimates the likelihood of a patient developing lung cancer within 1 to 6 years. Given the high correlation observed across all six risk estimates, we selected the 1-year risk score.

Longitudinal features: In contrast to a single “delta-radiomics” measurement where features are subtracted between only two time points, we leveraged time-varying survival regression to incorporate all available time points. Specifically, we utilized the lifelines 0.27.8 Python 3.8 package for time-varying survival regression, which allows each patient’s radiomics and six Sybil features to be tracked over multiple intervals throughout treatment and follow-up [9,10]. We employed time-varying survival regression to identify features with the highest hazard ratios (HRs). The significant HRs from these features were selected.

Demographic Characteristics: Smoking status data were obtained from patient records. Age and gender were also included as demographic factors.

### 2.4. Dynamic Lung Cancer Risk Modeling

To dynamically predict lung cancer risk, we used Random Survival Forests (RSFs) based on the extracted multi-modal features. Our dataset exhibited considerable heterogeneity in the number of CT scans per patient and the time intervals between these imaging studies. To address this, we adopted a nested cross-validation strategy, ensuring data was split at the patient level to prevent data leakage.

Specifically, patients were partitioned into training and testing groups in an outer split, using a 4:1 ratio. Given the variability in the number of CT scans per patient, the training and testing sets might have differences in the number of scans included. This splitting was repeated 100 times to evaluate the robustness of evaluated models, forming the outer loop of our nested cross-validation approach.

Within each iteration of the outer loop, we trained a Random Survival Forests (RSFs) model and optimized the model’s hyperparameters for each training cohort. We conducted a detailed grid search, embedded within a five-fold cross-validation procedure (inner loop), to explore a predefined set of hyperparameters. This process aims to identify the optimal model configuration. The model that demonstrated the highest performance on the inner validation cohort was then selected and evaluated on the outer test cohort, and its performance was reported accordingly.

### 2.5. Statistical Analysis

Harrell’s C-statistics (concordance index [C-index]) was used for feature selection and served as a primary metric for comparing models built on different feature sets. We employed Kaplan–Meier (KM) curves as a primary metric for model comparison. For feature variation analysis, *p*-values from the Kruskal–Wallis rank-sum test were used to determine whether the top-selected features differed significantly across various time intervals leading to cancer development. For KM curve comparisons, *p*-values from the log-rank test were used to evaluate whether there were statistically significant differences in risk curves.

## 3. Results

A total of 122 eligible patients were included in the analysis (Table 1). Among these individuals, 67 (55%) had a history of cancer, most commonly breast (8%), skin (7%), or blood cancer (7%). This cohort underwent a total of 622 CT scans (Figure 1a). The median number of CT scans per patient was 4.0 (interquartile range [IQR], 3.0–6.0). Patients’ first CT scans were performed a median of 1.35 (IQR 0.52–3.23) years before cancer diagnosis. The cohort had a mean age of 68.1 ± 11.1 years, with 36.2% identifying as male and 63.8% as female.

### 3.1. Radiomics Feature Preprocessing

For the radiomics and delta-radiomics features, the feature selection procedure is presented in Appendix A. We employed hierarchical clustering on these features, using the gap statistic [11] to determine the optimal number of clusters in each nested cross-validation. Subsequently, from each cluster, we selected the representative feature by selecting the one that yielded the highest C-index with Cox Proportional Hazards model [12].

The colormap for TtCD (time to cancer development), representing the clustered patient samples, reflected moderate clustering trends only for TtCD > 3 years. The t-SNE plot, representing all the features for three TtCD-based groups (<1 year, 1–3 years, and >3 years), is shown in Appendix A. While there are two somewhat distinct regions corresponding to TtCD > 3-years and < 1-year, most regions are mixed, highlighting the challenges in predicting cancer risk in the cohort.

### 3.2. Individual Radiomics Features Closely Associated with Lung Cancer Risk

Features chosen in over 50% of 100 nested CV iterations across different types of feature sets are displayed in Figure 2a. Within radiomic features, two morphological features including “surface volume ratio” and “least axis length” were top selected, followed by “mean,” “dependence entropy” and “LargeDependenceHighGrayLevelEmphasis.” For Dradiomic features, “surface area” was chosen in all iterations, with “run length non-uniformity normalized” following closely. Among longitudinal features, the Sybil feature “1-year-risk,” followed by the “surface volume ratio,” “3-year-risk” and “RootMeanSquared” were selected.

For the five top selected radiomic features, we observed that they had a clear trend of either increase or decrease based on the risk of lung cancer development (Figure 2b), as stratified in 4 groups including the “<1-year” group for those diagnosed within a year, the “1–3 year” group for patients diagnosed between one and three years, and for “>3-year” and “>7-year” groups for individuals diagnosed after more than three and seven years, respectively.

When CT images approached a cancer diagnosis, we noted a decrease in the “surface volume ratio” feature. Geometrically, this indicates that the volume of the nodule was growing faster than its surface area, often implying a bulkier, more sphere-like structure. While malignancies can develop localized surface irregularities (e.g., spiculation), their rapid volumetric expansion still reduces the overall surface-to-volume ratio.

Furthermore, the median “least axis length” for patients diagnosed with cancer within the next year was approximately 12 mm, compared to 6 mm for those diagnosed more than 3 years later. Additionally, there was a significant increase in the mean Hounsfield units (HU) as follows: 37% between the “>3-year” and “1–3 year” groups; 34% between the “1–3 year” and “<1-year” groups. This observation suggests that nodules become denser and more solid as the time to cancer diagnosis decreases. Furthermore, there was a rising trend in “dependence entropy,” which points to increasingly complex textural features of nodules as patients near the point of developing cancer. These changes in entropy, as captured in the textural features of nodules on CT images, could reflect the underlying heterogeneity of tumor microenvironments.

In the analysis of longitudinal features (Figure 2b(i,vi–ix)), significant variations were observed in “SurfaceVolumeRatio” and “root mean squared” features. Evaluation of Sybil features in longitudinal feature selection (1-year-risk and 3-year-risk) showed limited statistical significance at population level. This may suggest that while they may not generalize broadly, they could still provide useful information when examined within an individual’s serial CT scans. For the two DRadiomics features (Figure 2b(x,xi)), “Surface Area” demonstrated a significant increasing trend as patients approached cancer diagnosis, whereas “RunLengthNonUniformityNormalized” showed a bell-shaped trajectory, rising at first and then falling.

To highlight imaging, demographic, and longitudinal variables leading to the model’s cancer risk predictions, first we plotted a correlation matrix to illustrate the relationships between patient demographic characteristics and the most frequently selected features (Figure 3a). We observed minimal correlation between different feature types (demographic, radiomics, and Sybil), with few correlations observed within each feature group. Furthermore, we analyzed the individual features’ contributions (Figure 3b) of a representative model highlighted that “surface area”, “surface volume ratio”, and its longitudinal features, together with “entropy” and “least axis length”, are the top five most important features for prediction.

### 3.3. Composite Risk Model Achieves Optimal Dynamic Lung Cancer Risk Prediction

Figure 4a shows the median C-indices of different lung cancer risk models using different types of features. Employing only demographic features yielded a baseline C-index of 0.50, followed by Sybil with a C-index of 0.54. Radiomic features resulted in a median C-index of 0.66. We achieved the highest median C-index (0.69) when integrating all feature types: demographics, Sybil, radiomics, Dradiomics, and longitudinal features.

To assess model robustness, we compared the Random Survival Forest (RSF) with a Lasso-Cox model and observed higher concordance for RSF. As shown in Appendix A, RSF achieved consistently better performance, while Appendix A illustrates a modest increase in C-index with larger training sample sizes, suggesting improved model stability with more data.

We have further compared the performance of different models through KM curves and personalized cancer development probability functions (Figure 4b). In the KM plot, patients are divided into two risk groups (high and low risk) based on the median predicted risk values. We present scatter plots of the predicted probability across all CTs in the test set in Appendix A. A trend of increasing cancer probability with lower TtCP is observed in Appendix A. This pattern is also evident in the box plots for combined features (All) and radiomics, reinforcing the observed trend. Additionally, the predicted cancer development probability functions of six patients during tests using composite model were plotted, which offers a detailed view of how the model’s predictions evolve over time (Figure 4c).

We randomly selected five cases from the outer test cohort for visualization of predicted dynamic model risk side-by-side with nodule appearance on CT scans (Figure 5a). Scatter plots (Figure 5b) of cancer development probability versus time-to-cancer development (TtCD) demonstrate a relatively consistent trend: as TtCD decreases, the probability of cancer increases for both the composite model and the Brock Model. For the Brock model, using the 10% threshold [13], the results indicate 67% accuracy, 100% sensitivity, and 33% specificity. In comparison, the composite model, with a 10% cutoff, achieves 78% accuracy, 89% sensitivity, and 67% specificity.

## 4. Discussion

In this study, we conducted a comprehensive investigation of longitudinal radiomic features for early lung cancer risk prediction in patients ineligible for USPSTF screening, a major lung cancer population not included in most lung cancer screening trials and therefore not included in screening guidelines [14,15,16]. Despite the 2021 USPSTF revisions expanding screening eligibility, studies indicate that at least 30–50% of lung cancer patients remain ineligible for screening under the updated criteria [17,18]. We analyzed serial CT scans from a real-world population, capturing classical radiomic features (e.g., shape, texture) and associated temporal changes (∆radiomics). By fitting a time-varying survival model that integrated radiomics and demographic variables, it can dynamically predict individual patients’ future risk of developing lung cancer. Our approach, which utilizes time-dependent information from serial CT scans, may provide better predictive performance in individuals ineligible for USPSTF screening, as existing models like Brock and Sybil were designed for screening-eligible populations.

Future lung cancer risk modeling is essential for non-screening populations, particularly non-smokers and light smokers, who are often overlooked in traditional lung cancer screening programs despite accounting for a meaningful portion of lung cancer cases [19,20]. The TALENT trial in Taiwan demonstrated the effectiveness of LDCT screening in non-smokers and light smokers, reinforcing the need for tailored risk models in this population [21]. Unlike high-risk smokers eligible for routine low-dose CT scans, these individuals typically lack early detection opportunities, leading to delayed diagnoses and poorer outcomes [22]. Adenocarcinoma, the most common form of lung cancer in nonsmokers, typically originates in the peripheral regions of the lungs. These tumors tend to grow more slowly and are less likely to metastasize in the early stages compared to other lung cancers, making early detection particularly beneficial. Identifying high-risk nodules at an early stage offers a crucial window for interception, which can significantly improve treatment outcomes and patient survival [23]. Timely diagnosis and management of lung cancer in nonsmokers are vital, as early-stage detection has the potential to enhance therapeutic effectiveness [24]. By integrating radiomic features and time-varying survival models, future risk prediction frameworks can dynamically assess individual cancer risk trajectories, enabling the early identification of high-risk nodules before they become clinically evident. Furthermore, these risk prediction models can play a crucial role in guiding patient enrollment for immunoprevention trials, such as IMPRINT-Lung (NCT03634241) and Can-Prevent-Lung (NCT04789681) [25], which aim to intercept malignant progression at its earliest stages.

Our study supports prior radiomics studies [26,27] by identifying specific CT patterns that offer critical insights into malignant transformation, distinct from prior deep learning approaches [4,28] which function as “black-box” models where the decision-making process and the individual features contributing to predictions are not easily interpretable. Besides nodule size, the significant increase in CT voxel intensity as the time to diagnosis shortens reflects the transition from ground-glass to solid nodules, indicating increased cellular density, reduced aeration, and fibrotic remodeling—key hallmarks of malignant progression. These features align closely with existing Lung-RADS criteria [29], but offer a more quantitative, three-dimensional assessment. Additionally, we observed a decrease in the surface-to-volume ratio, suggesting that malignant nodules expand in volume more rapidly than in surface area, a characteristic of aggressive tumor growth. Moreover, the rising trend in dependence entropy highlights the increasing heterogeneity and structural complexity of nodules approaching diagnosis, reinforcing the correlation between intratumoral angiogenesis, heterogeneity, and more aggressive tumor behavior with poorer clinical outcomes. These findings underscore the clinical relevance of radiomic features in tracking lung cancer evolution, supporting their integration into predictive models for early risk assessment and intervention planning.

Interestingly, we observed an enhanced predictive effect when combining nodule-based radiomic patterns with the deep learning-based whole-lung Sybil model, suggesting that integrating complementary features may enhance predictive accuracy and improve early lung cancer risk assessment. Additionally, this indicates the possible presence of previously unrecognized CT imaging patterns that may be associated with nodule risk, highlighting the need for further exploration of novel radiomic biomarkers to improve early lung cancer detection and risk stratification. Furthermore, we recognize that the small, monocentric nature of our cohort limits immediate generalizability and necessitates external validation in independent cohorts. In addition, integrating radiomic features with molecular data [30], such as circulating tumor DNA (ctDNA) and proteomic profiling, could enhance our understanding of lung cancer risk and facilitate the development of multi-modal predictive models. In addition, mechanistic studies, integrating imaging, histopathology, and molecular profiling, are needed to establish the biological basis of these radiomic features and their role in tumor development. Also, it would be valuable to compare these evolution patterns with those observed in USPSTF-eligible populations, providing valuable insights into risk divergence between the two groups.

## 5. Conclusions

In a real-world cohort of patients ineligible for USPSTF screening, we showed that longitudinal CT information contains actionable signals for future lung cancer risk. By tracking how nodules and surrounding lung parenchyma evolve across serial scans, converting those changes into quantitative radiomics, and modeling them with time-varying survival methods, our composite approach improved risk discrimination over conventional tools. Relative to demographics alone, a single-scan deep model, and the Brock model, our method delivered higher accuracy and specificity while maintaining high sensitivity, offering clearer risk separation for non- and light-smokers.

Our findings point to a practical pathway for earlier evaluation and intervention in USPSTF-ineligible patients: leverage serial CT scans to quantify longitudinal nodule and lung features (radiomics, Δradiomics, and Sybil-derived signals), integrate these with basic clinical factors via time-varying survival modeling, and produce personalized cancer development probability functions to guide follow-up timing and intensity. The visual correspondence between rising predicted risk and evolving nodule characteristics on longitudinal CT, such as increasing size, density, entropy, and decreasing surface-to-volume ratio, supports the face validity and clinical utility of this dynamic risk modeling approach.

This single-center, retrospective study had a modest sample size and relied on manual segmentations across heterogeneous scan protocols (contrast and non-contrast). Importantly, more than half of the cohort had a prior cancer history (55%), which may have conferred a higher baseline risk than populations without prior cancer and could also have affected generalizability and risk calibration. External validation, prospective testing, automated and robust nodule segmentation, and protocol harmonization are needed before deployment. Future work will also explore integrating molecular markers and broader whole-lung features to build multimodal, generalizable models for USPSTF-ineligible populations.

## Figures and Tables

**Figure 1 cancers-17-03406-f001:**
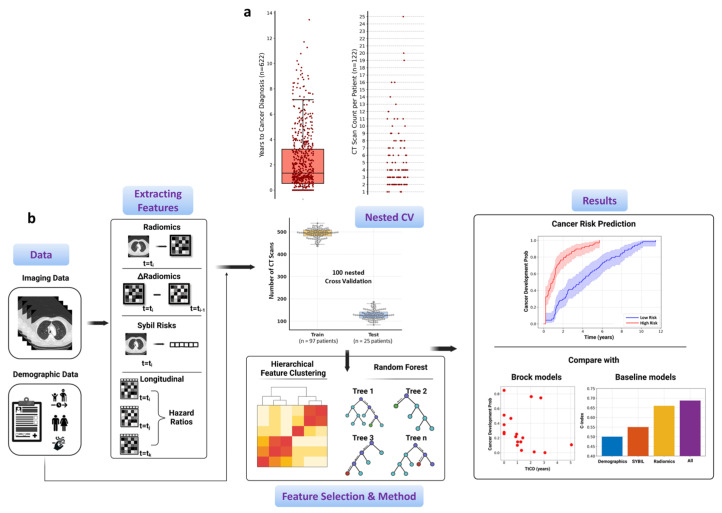
Overview of the distributions of time to cancer development, CT scan frequencies, and feature extraction for cancer risk prediction in our nested cross-validation approach. (**a**) Distribution of time to cancer diagnosis, where the median time to cancer development is 1.4 years, and CT scan frequencies per patient, showing that most patients have 2, 3, or 4 scans before diagnosis. (**b**) Workflow summarizing feature extraction, feature selection, and downstream modeling steps for risk prediction under nested cross-validation.

**Figure 2 cancers-17-03406-f002:**
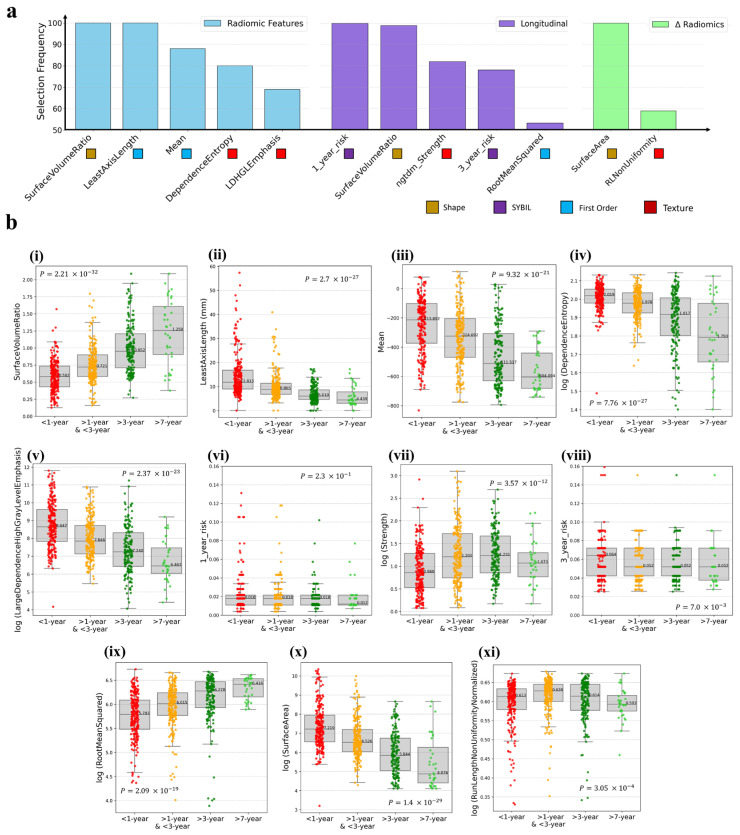
Selected features with >50% frequency in hierarchical clustering across 100 nested CV iteration and their variations over different time intervals leading to cancer development (TtCD). (**a**) Frequency selection of radiomic, radiomic differences, and longitudinal features. (**b**) Box plots display these frequently selected features over different TtCD intervals. The box plots categorize patients into four distinct groups: those who developed lung cancer within one year, those who developed it between one and three years, those who developed it after more than three years, and those diagnosed more than seven years after the initial observation.

**Figure 3 cancers-17-03406-f003:**
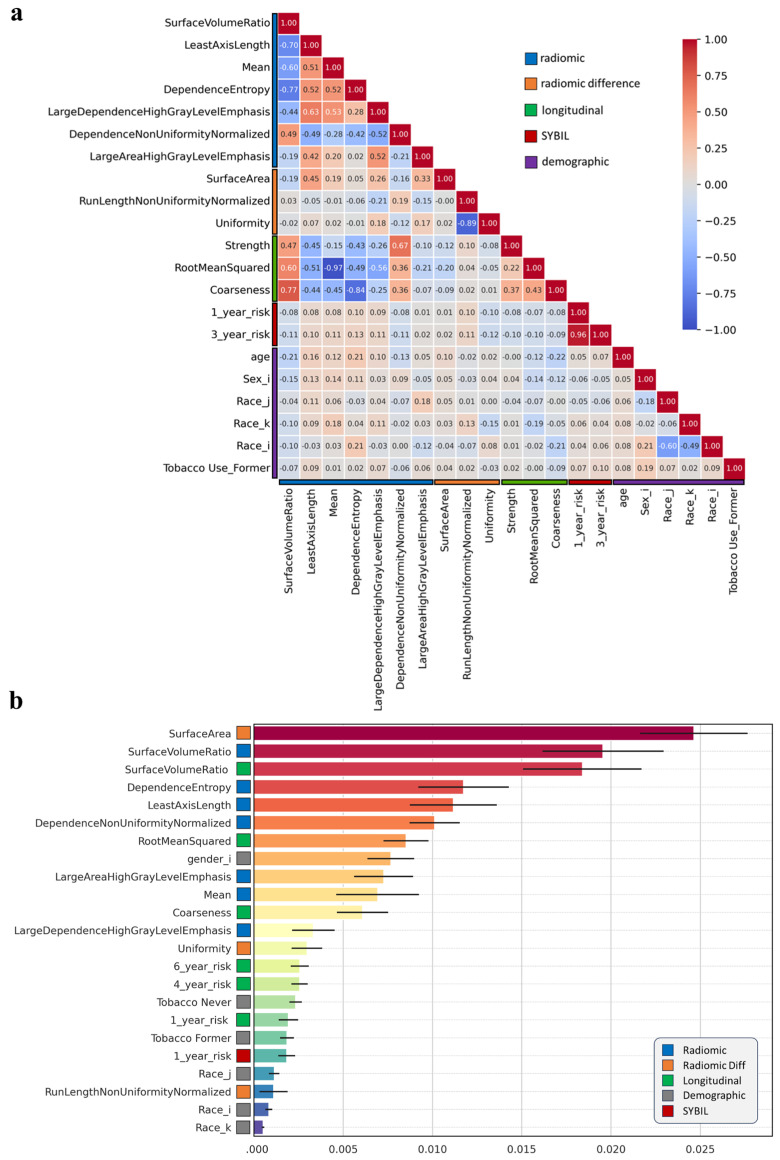
Selected Feature relationships and contributions to cancer prediction. (**a**) Correlation matrix illustrating the relationships between the most frequently selected features across different feature sets, as well as demographic features. (**b**) Contribution of each feature in predicting cancer development, ranked by significance and categorized by type, for a representative model using combined features. Noting that the features, characterized by longitudinal ones, were employed, and evaluated based on their log hazard ratios.

**Figure 4 cancers-17-03406-f004:**
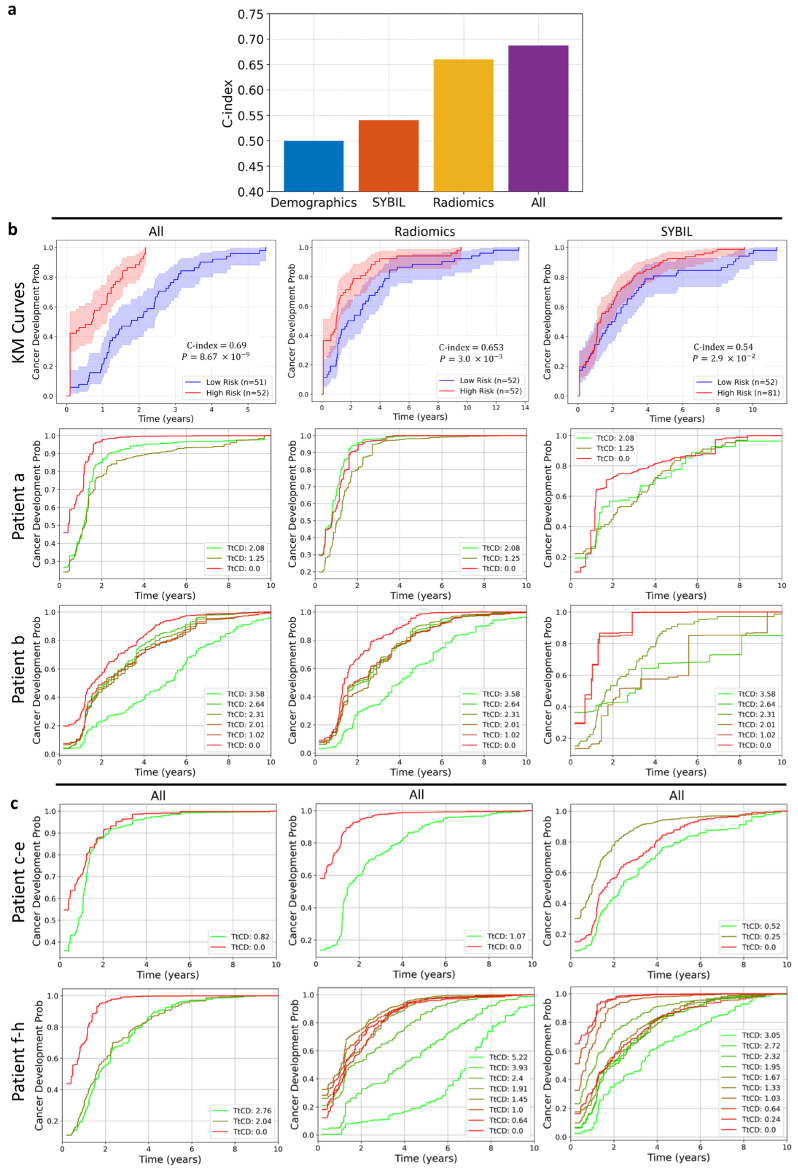
Evaluation of predictive models for cancer development using nested CV and personalized risk estimation. (**a**) Evaluating the predictive power of different feature sets; demographics, Sybil, radiomics, and all combined (demographics, Sybil, radiomics, radiomic difference, and longitudinal risks) on cancer development using a nest-ed cross-validation approach (n = 100). The median C-index values are shown for the patients’ CT scans in the outer test cohort. (**b**) Kaplan–Meier (KM) curves and personalized cancer development probability functions for each model. The KM curves illustrate cancer development in patients’ CT scans, categorized by each model based on feature sets. Below each KM curve, probability curves for two selected patients (selected from the test set, common across all models, to prevent information leakage) depict how predicted cancer risk evolves over time as the patient approaches the diagnosis date. The color transition from green to red represents predictions from the CT scan furthest from to the one closest to diagnosis. (**c**) The personalized cancer development probability function for six patients using all features in the outer test cohort. In the figure legends, TtCP denotes the time to cancer development, with the numerical value preceding it indicating the time in years.

**Figure 5 cancers-17-03406-f005:**
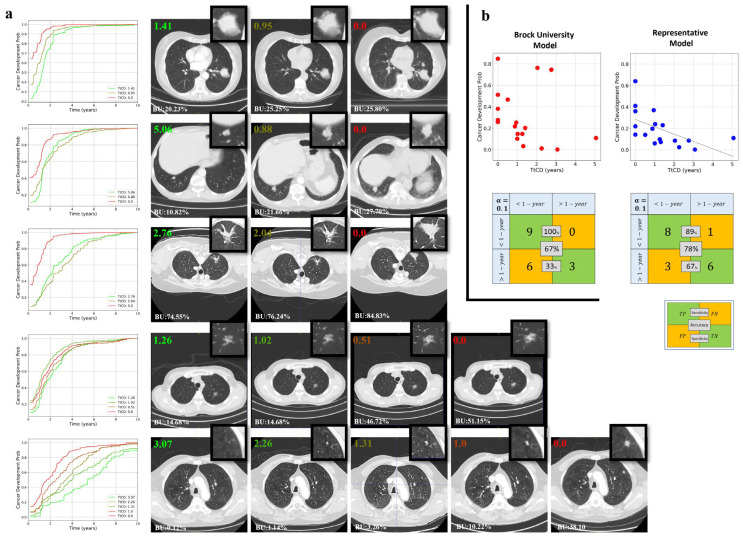
Comparative visualization of cancer development probability in longitudinal CT scans for five patients in the outer test cohort of the representative model incorporating all features. (**a**) This Figure displays personalized cancer development probability functions for five, each represented by 3, 3, 3, 4, and 5 longitudinal CT scans, respectively. The leftmost image in each row shows the obtained cancer development probability functions. In each CT scan, the number in the upper left corner indicates the remaining time to cancer development (TtCD), the box in the upper right corner shows the magnitude of the nodule area, and the number in the lower left corner provides the Brock University risk score. (**b**) Scatter plots of cancer development probability vs. TtCD for our representative model and the Brock University model (Model 2b) [3], along with the confusion matrix for both models using a 10% cutoff to categorize patients into “<1-year” and “>1-year” groups.

**Table 1 cancers-17-03406-t001:** Demographics information of 122 patients with 622 CT scans.

Parameter	Patients (n = 122)	CTs (n = 622)
Sex		
Female	75 (61%)	397 (64%)
Male	47 (39%)	225 (36%)
Race		
White	102 (84%)	512 (82%)
Black	9 (7%)	45 (7%)
Asian	6 (5%)	34 (5%)
Other	5 (4%)	31 (5%)
Tobacco Use		
Former	84 (69%)	375 (60%)
--Former (<20 pack-year history)	39 (32%)	
--Former (quit >15 years ago)	45 (37%)	
Never (<100 lifetime cigarettes)	36 (30%)	209 (34%)
Current	2 (2%)	38 (6%)
Imaging Type		
Contrast CT		280 (45%)
Non-Contrast CT		342 (55%)
Prior Cancer History		
Breast Cancer	10 (8%)	
Skin Cancer	9 (7%)	
Blood Cancer	8 (7%)	
Head and Neck	7 (6%)	
Other Cancer	33 (27%)	
No Cancer History	55 (45%)	
Time to Cancer Development (TtDC)		Med: 1.4 Min: 0.0 Max: 13.5
Age for patients (n = 122), represented at the time of cancer diagnosis	Med: 72.62 Min: 42.79 Max: 89.19	Med: 70.67 Min: 34.05 Max: 89.19

## Data Availability

The data used in this study will be accessible to the research community upon reasonable request from the corresponding author.

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
