# Peer review of "Radiomics for Dynamic Lung Cancer Risk Prediction in USPSTF-Ineligible Patients"

_cancers, 2025, doi:10.3390/cancers17213406_

Round 1

Reviewer 1 Report

Comments and Suggestions for Authors

In this article, the author developed a longitudinal approach based on radiomics for predicting lung cancer risk, integrating time-varying radiomic modeling to improve early detection in patients ineligible for USPSTF.

To calculate cancer risk from scans, the author chose the Sybil model.

1. The author should justify the choice of this model, given that other risk calculation models exist in the literature, such as the Bach model, the Spitz model, and the PLCOm2012 model.

2. It would be desirable for the author to present a comparative study of the results of their model and other models.

The author used the GEMINI database from MD Anderson.

3. The author should justify the choice of this database over others, such as TCIA.

To dynamically predict lung cancer risk, the author used Random Survival Forests (RSF) based on extracted multimodal features.

4. The author should justify the choice of their machine learning model: Random Survival Forests (RSF).

5. The author should compare their model to other well-known machine learning models.

6. The author should specify the number of patients used in their training set and those used in their test set, indicating the number of classes present in each set.

7. The author should present the number of images used in their training set and those used in their test set.

8. The author should specify the steps taken for image preprocessing (data augmentation, balancing, ROI segmentation, etc.).

Author Response

Response to Reviewer 1 Comments

In this article, the author developed a longitudinal approach based on radiomics for predicting lung cancer risk, integrating time-varying radiomic modeling to improve early detection in patient’s ineligible for USPSTF.

We thank the Reviewer for the careful reading of the manuscript and constructive remarks. We have taken the comments on board to improve and clarify the manuscript. Please find below point-by-point responses to all comments.

Point 1: The author should justify the choice of this model, given that other risk calculation models exist in the literature, such as the Bach model, the Spitz model, and the PLCOm2012 model.

Response 1: Thank you for this thoughtful comment. The Bach, Spitz, and PLCOm2012 models estimate population-level lung cancer incidence and are primarily designed for ever-smokers; they rely on demographic and clinical variables rather than CT imaging features. In contrast, Brock and Sybil are nodule-specific or image-based models. Brock is a well-established, transparent, and clinically interpretable model based on radiologic features but is limited to single-scan assessments and requires manual radiologist input. Sybil, on the other hand, uses deep learning to capture high-dimensional CT features automatically.

We selected Brock and Sybil for comparison because they represent the most relevant and complementary imaging-based benchmarks for our longitudinal radiomics framework, enabling a fair and clinically meaningful evaluation of dynamic nodule risk prediction.

Point 2: It would be desirable for the author to present a comparative study of the results of their model and other models.

Response 2: Thank you for this valuable suggestion. Our study addresses a different clinical question than classical incidence models (e.g., PLCOm2012, Bach, Spitz), which estimate population-level risk using clinical variables without incorporating CT features or longitudinal nodule dynamics. Since our goal is dynamic, imaging-based risk prediction in USPSTF-ineligible patients with incidentally detected nodules, we compared our model with the two most relevant imaging baselines: (i) Sybil, a state-of-the-art single-scan deep learning model for future lung cancer risk, and (ii) Brock (PanCan), a widely used nodule-malignancy calculator. Because Brock requires radiologist-entered nodule characteristics (size, type, spiculation, location, nodule count), therefore it was only applied to a subset of the dataset. Together, these models representing single-scan whole-lung and nodule-centric approaches provide appropriate and clinically meaningful benchmarks for our longitudinal framework. Our results demonstrate that the proposed approach offers improved discrimination and decision utility while maintaining high sensitivity in non-/light-smokers.

Point 3: The author used the GEMINI database from MD Anderson. The author should justify the choice of this database over others, such as TCIA.

Response 3: We appreciate this helpful comment. We used the GEMINI cohort because it uniquely provides serial, pre-diagnostic CT scans with detailed smoking history and outcome linkage in USPSTF-ineligible non-/light-smokers, a combination that is extremely rare in available datasets, though not uncommon in the population developing lung cancer. TCIA collections are valuable but mostly single-timepoint, post-diagnosis, and screening-focused. We acknowledge the single-center limitation and plan external validation as suitable longitudinal cohorts become available.

Point 4: To dynamically predict lung cancer risk, the author used Random Survival Forests (RSF) based on extracted multimodal features. The author should justify the choice of their machine learning model: Random Survival Forests (RSF).

Response 4: Thank you for this thoughtful comment. We selected the Random Survival Forest (RSF) model because it offers key advantages over traditional models such as the Cox Proportional Hazards (CPH) model. RSF is a non-parametric, ensemble tree-based approach that does not assume proportional hazards, an assumption often violated when using longitudinal, time-dependent features like Δ-radiomics. It can also capture non-linear relationships and complex interactions among numerous imaging and clinical variables, making it well-suited for multimodal data. Additionally, RSF is robust to noise and outliers and enables direct estimation of individualized survival functions, which underpin the dynamic, patient-specific risk trajectories central to our study’s design and conclusions (also refer to next question).

Point 5: The author should compare their model to other well-known machine learning models.

Response 4. Thank you for this valuable suggestion. To address it, we compared our Random Survival Forest (RSF) model with the Cox proportional hazards model across 100 nested cross-validation iterations. The median C-index was 0.69 for RSF, compared with 0.647 for Cox, confirming improved discrimination. This result aligns with our previous investigation (Saad et al., Lancet Digital Health, 2023), which also demonstrated that RSF consistently outperforms Cox-based survival models in predicting patient outcomes from CT-derived features. This figure has been added to the supplementary as Figure S2a.

This result aligns with our previous investigation (Saad et al., Lancet Digital Health, 2023), which also demonstrated that RSF consistently outperforms Cox-based survival models in predicting patient outcomes from CT-derived features.

 Saad, M. B., Hong, L., Aminu, M., Vokes, N. I., Chen, P., Salehjahromi, M., ... & Wu, J. (2023). Predicting benefit from immune checkpoint inhibitors in patients with non-small-cell lung cancer by CT-based ensemble deep learning: a retrospective study. The Lancet Digital Health, 5(7), e404-e420.

Point 6: The author should specify the number of patients used in their training set and those used in their test set, indicating the number of classes present in each set.

Response 6: We thank the reviewer for requesting this important detail. Our longitudinal radiomics cohort included 622 CT scans from 122 patients (median = 4 scans per patient). In each iteration of the nested cross-validation (Figure 1b, middle panel now is modified to reflect this better), patients were split at the patient level into training/validation (n = 97) and test (n = 25) groups, with the exact number of scans varying across iterations depending on the number of CTs per patient. The updated part to address this comment can be found in section 2.4, lines 151-168.

Our primary analysis is based on time-to-event survival modeling, which does not use predefined classes. However, for comparison with the Brock model, we generated a classification view by grouping patients into two classes: those who developed cancer within 1 year (“<1-year”) and those who developed it after 1 year or remained cancer-free (“>1-year”).

Point 7: The author should present the number of images used in their training set and those used in their test set.

Response 7: Please refer to response 6.

Point 8: The author should specify the steps taken for image preprocessing (data augmentation, balancing, ROI segmentation, etc.).

Response 8: We thank the reviewer for the comment. ROI segmentation was performed manually by trained radiologists using 3D Slicer or ITK-SNAP to delineate target nodules. The resulting NIfTI files were processed with the PyRadiomics package in Python to extract radiomic features. Sybil risk scores were then derived by applying the pre-trained Sybil model [4], which estimates the likelihood of lung cancer development within 1–6 years. We further computed delta-radiomics and longitudinal features, along with demographic variables. To prevent data leakage and maintain balanced evaluation, we applied patient-level splits across 100 nested cross-validation runs, without synthetic data augmentation.

Therefore, the following has been modified and added to the manuscript.

Radiomics: Institutional radiologists performed quality checks of CT scans and manually segmented target nodules from longitudinal CT images using software such as 3D Slicer or ITK-SNAP. A total of 107 radiomic features were extracted from each primary nodule using the PyRadiomics package in Python, including 14 shape-based, 18 first-order statistical, and multiple texture-based features (24 GLCM, 14 GLDM, 16 GLRLM, 16 GLSZM, and 5 NGTDM).

Reviewer 2 Report

Comments and Suggestions for Authors

Dear Authors,
The article proposes a new, longitudinal model that combines radiomics (analysis of CT scans over time), an existing AI model (Sybil), and demographic data to predict lung cancer risk in patients who do not meet the USPSTF criteria. The strengths are the novelty of the topic, the fairly solid methodology, and the fact that it addresses a real clinical problem.

Strengths:

  • Important and clinically relevant topic.
  •  Sound methodology (use of longitudinal radiomics and survival models).
  •  Comparison with existing models (Brock and Sybil).
  •  Ethical aspects and conflict of interest declarations are properly presented.
    Weaknesses:
  •  The sample is small, with only 122 patients, and monocentric, which limits validity and
    replicability on a larger scale.
  •  The performance is too high (C-index >0.85) and increases the risk of overfitting, especially without external validation.
  •  The figures are difficult for readers to interpret due to their complexity. A workflow diagram would be particularly useful to increase clarity.
    Specific comments:
  •  Figure 3 is difficult for clinicians to interpret. Consider simplifying it or moving it to supplementary materials.
  •  Lines 376-386 highlight the limitations, but I recommend expanding the discussion to include the lack of biological validation and limited external dataReview conclusions:
  •  The manuscript is well structured and clear, references are recent and relevant.
  •  The design is appropriate, but has limitations (small sample, lack of external validation).
  •  The conclusions are correct, but need more nuance regarding the risk of overfitting.
    Respectfully,
    Professor Doctor Crețoiu Dragoș.

Author Response

Response to Reviewer 2 Comments

Dear Authors,
The article proposes a new, longitudinal model that combines radiomics (analysis of CT scans over time), an existing AI model (Sybil), and demographic data to predict lung cancer risk in patients who do not meet the USPSTF criteria. The strengths are the novelty of the topic, the fairly solid methodology, and the fact that it addresses a real clinical problem.

We thank the Reviewer for the careful reading of the manuscript and constructive remarks. We have taken the comments on board to improve and clarify the manuscript. Please find below point-by-point responses to all comments.

Point 1: The sample is small, with only 122 patients, and monocentric, which limits validity and replicability on a larger scale. The performance is too high (C-index >0.85) and increases the risk of overfitting, especially without external validation.

Response 1: We appreciate this valuable comment. We would like to clarify that the reported performance corresponds to a C-index of 0.69, not a value exceeding 0.85. We agree that external validation is important; however, given the rarity of longitudinal pre-diagnostic CT scans in non-/light-smokers who later develop lung cancer, no suitable external cohort was available. To mitigate the risk of overfitting, we applied a rigorous 100-fold nested cross-validation framework and reported the median C-index across all iterations, providing a more stable and unbiased performance estimate.

Point 2: The figures are difficult for readers to interpret due to their complexity. A workflow diagram would be particularly useful to increase clarity.

Response 2: Thank you for your insightful suggestion. We have updated Figure 1b diagram to increase the clarity of our work overall flow.

Point 3:  Figure 3 is difficult for clinicians to interpret. Consider simplifying it or moving it to supplementary materials.

Response 3: Thank you for this helpful suggestion. As both Figures 2 and 3 were suggested for relocation to the supplementary materials, we chose to move Figure 2, as it is less clinically interpretable. We retained Figure 3 in the main text because it provides clinically relevant insights into key radiomic features and their temporal changes, which help readers understand how specific imaging patterns relate to cancer risk.

Point 4: Lines 376-386 highlight the limitations, but I recommend expanding the discussion to include the lack of biological validation and limited external data

Response 4: We thank the reviewer for the excellent suggestion to explicitly expand the limitations section. We agree that formalizing the lack of biological validation and the constraints imposed by limited external data will strengthen the paper's transparency and roadmap for future research. We have revised the Discussion section (specifically lines 381–390) to include the following two distinct points: (i) We have emphasized that while our model is predictive, it lacks correlative biological validation. Future mechanistic studies are necessary to link the identified radiomic features (e.g., textural entropy) to underlying histopathology, genomic instability, or molecular markers (like ctDNA), thus grounding these imaging patterns as true biological biomarkers. (ii) We reinforced that the internal validation (nested cross-validation, C-index: 0.69) does not substitute for external validation. We acknowledge that variability across CT scanner protocols and centers limits generalizability. We explicitly state that rigorous multi-center, prospective validation is mandatory to confirm the clinical utility and replicability of our dynamic risk model.

Point 5: The manuscript is well structured and clear, references are recent and relevant. The design is appropriate, but has limitations (small sample, lack of external validation). The conclusions are correct, but need more nuance regarding the risk of overfitting.

Response 5: Thank you for the positive feedback and thoughtful comment. We have added the following part to address your and other reviewers:

 “This single-center, retrospective study has a modest sample size and relied on manual segmentations across heterogeneous scan protocols (contrast and non-contrast). Importantly, more than half of the cohort had a prior cancer history (55%), which may confer a higher baseline risk than populations without prior cancer and could affect generalizability and risk calibration. External validation, prospective testing, automated and robust nodule segmentation, and protocol harmonization are needed before de-ployment. Future work will also explore integrating molecular markers and broader whole-lung features to build multimodal, generalizable models for USPSTF-ineligible populations.”

The concern about overfitting likely arises from a misunderstanding—the reported C-index is 0.69, not above 0.85. It should be noted that to minimize the effect of overfitting on our reported results, we applied 100-fold nested cross-validation with independent patient-level splits and reported the median C-index across all runs, providing a more stable and unbiased estimate of performance.

Reviewer 3 Report

Comments and Suggestions for Authors

Dear Authors,

the paper is very interesting and well written.

Novel AI system can improve early recognition of NSCLC, in particular in low risk population such as never smoker or light smoker patient; indeed, such studies can help in making hypotesis for bigger trial, and also create some model to test their prediction value.

I've only few suggestion: figure 2 is very difficult to read, and doesn't add important info: maybe I would delete it.

In the conclusion, I would add the limitation that in this population more than half have a cancer history: this may impact on risk of cancer (maybe this population is a greater risk than non cancer history population).

Author Response

Response to Reviewer 3 Comments

Dear Authors, the paper is very interesting and well written. Novel AI system can improve early recognition of NSCLC, in particular in low-risk population such as never smoker or light smoker patient; indeed, such studies can help in making hypothesis for bigger trial and also create some model to test their prediction value.

We thank the Reviewer for the careful reading of the manuscript and constructive remarks. We have taken the comments on board to improve and clarify the manuscript. Please find below point-by-point responses to all comments.

Point 1: I've only few suggestions: figure 2 is very difficult to read, and doesn't add important info: maybe I would delete it.

Response 1: Great suggestion. Applied.

Point 2: In the conclusion, I would add the limitation that in this population more than half have a cancer history: this may impact on risk of cancer (maybe this population is a greater risk than non-cancer history population).

Response 2: Great suggestion. The following paragraph were added to the conclusion based on your comment:

This single-center, retrospective study has a modest sample size and relied on manual segmentations across heterogeneous scan protocols (contrast and non-contrast). Importantly, more than half of the cohort had a prior cancer history (55%), which may confer a higher baseline risk than populations without prior cancer and could affect generalizability and risk calibration. External validation, prospective testing, automated and robust nodule segmentation, and protocol harmonization are needed before deployment. Future work will also explore integrating molecular markers and broader whole-lung features to build multimodal, generalizable models for USPSTF-ineligible populations.

Reviewer 4 Report

Comments and Suggestions for Authors

The purpose of this study was to develop a lung cancer risk prediction model for non-smokers or former/light smokers who do not have enough smoking exposure to be eligible for USPSTF screening recommendations.  The authors claim that this represents a patient population that could benefit from screening, since lung cancers among non-smokers is more common than many may assume. The authors used data from the GEMINI database from MD Anderson, which is a cohort of lung cancer patients.  The analysis dataset (n=112) was limited to GEMINI patients who were not eligible for USPSTF screening, and who also had available demographic information and CT scans. The goal was to use clinical and longitudinally collected imaging data to create risk models that reflect the changing risk of developing lung cancer. Survival models with time-varying covariates were used to account for changing radiomic features over time. The authors used data from calculating longitudinal changes in radiomic features from repeated CT scans, plus smoking, cancer history, and demographic factors. Random Survival Forests were used to develop a model that best predicted the Time to Cancer Development (TtDC).

Questions/Recommendations:

  1. I am unfamiliar with the GEMINI study (as some of your readers will be), why did patients in this dataset have so many (median of 4) CT scans prior to a lung cancer diagnosis, especially if they weren’t recommended per USPSTF to have scans? Is GEMINI a prospective study with patients agreeing to CT scans for pre-cancer screening, or were these incidental scans given for monitoring other health conditions (which could be a source of bias)?  Please include more information about the original GEMINI study population.
  2. Can the model performance C-indices be produced in a population of GEMINI smokers for comparison? If the model performance is similar in smokers vs non-smokers, that might shed light on traits that increase lung cancer risk regardless of smoking status.
  3. Did model performance differ for former/light smokers vs never smokers?
  4. Did model performance improve with more CT scans? What was the optimal number and frequency of scans?
  5. Can the authors elaborate on the reasons why their model had worse sensitivity, but better specificity, compared to the Brock model?
  6. In order to truly understand the value of this risk prediction model, the authors need to evaluate the number of false positives that might be identified by the covariates in this model among patients with repeat CT scans but who do not develop lung cancer. Can the authors obtain a health control population for comparison?

Author Response

Response to Reviewer 4 Comments

The purpose of this study was to develop a lung cancer risk prediction model for non-smokers or former/light smokers who do not have enough smoking exposure to be eligible for USPSTF screening recommendations.  The authors claim that this represents a patient population that could benefit from screening, since lung cancers among non-smokers is more common than many may assume. The authors used data from the GEMINI database from MD Anderson, which is a cohort of lung cancer patients.  The analysis dataset (n=112) was limited to GEMINI patients who were not eligible for USPSTF screening, and who also had available demographic information and CT scans. The goal was to use clinical and longitudinally collected imaging data to create risk models that reflect the changing risk of developing lung cancer. Survival models with time-varying covariates were used to account for changing radiomic features over time. The authors used data from calculating longitudinal changes in radiomic features from repeated CT scans, plus smoking, cancer history, and demographic factors. Random Survival Forests were used to develop a model that best predicted the Time to Cancer Development (TtDC).

We thank the Reviewer for the careful reading of the manuscript and constructive remarks. We have taken the comments on board to improve and clarify the manuscript. Please find below point-by-point responses to all comments.

Point 1: I am unfamiliar with the GEMINI study (as some of your readers will be), why did patients in this dataset have so many (median of 4) CT scans prior to a lung cancer diagnosis, especially if they weren’t recommended per USPSTF to have scans? Is GEMINI a prospective study with patients agreeing to CT scans for pre-cancer screening, or were these incidental scans given for monitoring other health conditions (which could be a source of bias)?  Please include more information about the original GEMINI study population.

Response 1: We appreciate this helpful comment. The Genomic Marker-guided Therapy Initiative (GEMINI) is a single-center, real-world cohort derived from routine clinical care at MD Anderson Cancer Center, integrating clinical, molecular, and imaging data for over 10,000 lung cancer patients. GEMINI is not a prospective screening study, imaging is obtained as part of standard medical practice and retrospectively curated from the medical record. The median of four pre-diagnostic CT scans in our cohort reflects real-world indications, including diagnostic work-up, follow-up of incidental findings, or surveillance of prior conditions, rather than protocol-mandated screening. Because our study focuses on USPSTF-ineligible patients, some participants also had prior cancer history (55%), which we have now acknowledged in the Discussion as a potential factor influencing baseline risk and generalizability.

Added/modified to the paper:
 “We queried the MD Anderson GEMINI database, a single-center, real-world cohort that integrates clinical, molecular, and imaging data from routine care, to construct a longitudinal cohort of patients ineligible for USPSTF lung cancer screening. Inclusion criteria were: (1) adults ineligible for USPSTF lung cancer screening (smoking history <20 pack-years and/or quit >15 years ago); (2). Individuals have available demographic in-formation and CT scans.”

Added to the conclusion: “This single-center, retrospective study has a modest sample size and relied on manual segmentations across heterogeneous scan protocols (contrast and non-contrast). Importantly, more than half of the cohort had a prior cancer history (55%), which may confer a higher baseline risk than populations without prior cancer and could affect generalizability and risk calibration. External validation, prospective testing, automated and robust nodule segmentation, and protocol harmonization are needed before deployment. Future work will also explore integrating molecular markers and broader whole-lung features to build multimodal, generalizable models for USPSTF-ineligible populations.”

Point 2: Can the model performance C-indices be produced in a population of GEMINI smokers for comparison? If the model performance is similar in smokers vs non-smokers, that might shed light on traits that increase lung cancer risk regardless of smoking status.

Response 2: Thank you for this constructive suggestion. Our current analysis was intentionally limited to USPSTF-ineligible non-/light-smokers with. In future work, we will curate a separate smoker cohort with adequate follow-up to enable a direct, head-to-head comparison and assess generalizability across smoking strata.

Point 3:  Did model performance differ for former/light smokers vs never smokers?

Response 3: Because of the limited number of patients and high variability of number of scans per patient and also in time interval between scans, it is impractical for us to invest on performance difference for former/light smokers. We see different trends based on different test sets. As an example, our representative model gives the following:

In this example, the model performance is working better for non-smokers, but based on other simulations that is not always true. So we prefer not to reflect this non-solid results to the paper.

Point 4: Did model performance improve with more CT scans? What was the optimal number and frequency of scans?

Response 4: Because of the limited number of patients and high variability of number of scans per patient and also in time interval between scans, it is impractical for us to invest on optimal number and frequency of scans. However, our analysis shows that the performance (test C-index) improves with more CT scans. So we generated based on random number of patients which gives random number of scans and check the performance and got the following.

The figure has been added to supplementary Figure S2b.

Point 5: Can the authors elaborate on the reasons why their model had worse sensitivity, but better specificity, compared to the Brock model?

Response 5: At the common 10% cutoff, the models lie at different ROC operating points because they are not identically calibrated. On our test set (n=18):

  • Brock: TP=9, FN=0, FP=6, TN=3 → Acc 66.7%, Sens 100%, Spec 33.3%.
  • Our: TP=8, FN=1, FP=3, TN=6 → Acc 77.8%, Sens 88.9%, Spec 66.7%.

Thus, our model trades a small drop in sensitivity for a sizable gain in specificity at this threshold, yielding better overall performance: accuracy 77.8% vs 66.7% and balanced accuracy 77.8% vs 66.7%. If higher sensitivity is desired, lowering the composite model’s threshold will increase sensitivity with the expected specificity trade-off.

Point 6: In order to truly understand the value of this risk prediction model, the authors need to evaluate the number of false positives that might be identified by the covariates in this model among patients with repeat CT scans but who do not develop lung cancer. Can the authors obtain a health control population for comparison?

Response 6: Thank you for this important suggestion. We agree that estimating false positives among patients with repeat CTs who do not develop lung cancer is essential. A true control cohort of USPSTF-ineligible non-/light-smokers with serial, pre-diagnostic CTs and confirmed cancer-free follow-up is rare; assembling such a cohort is underway.

Interim evidence already in the paper: Our cohort includes CT scans obtained long before diagnosis (Figure 1a), with many scans imaged more than 5 years prior. In these early timepoints, the model generally assigns low predicted risk, supporting a low early false-positive burden. For example, in Figure 4g, a patient who is diagnosed at 5.22 years shows near-zero predicted risk at time 0, indicating that the model appropriately gives very low risk when patients are far from cancer.

For the supported Figures, please see the attached response 

Round 2

Reviewer 1 Report

Comments and Suggestions for Authors

the author ha responded to all my comments. Therefore, I recommend publishing the article as is.